# On Gibbs Energy for the Metastable bcc_A2 Phase with a Thermal Vacancy in Metals and Alloys

**DOI:** 10.3390/ma12020292

**Published:** 2019-01-17

**Authors:** Ying Tang, Lijun Zhang

**Affiliations:** 1School of Materials Science and Engineering, Hebei University of Technology, Tianjin 300130, China; 2State Key Laboratory of Powder Metallurgy, Central South University, Changsha, Hunan 410083, China

**Keywords:** thermal vacancy, metastable bcc_A2 phase, Gibbs energy, Ni-Zn system

## Abstract

An approach was proposed to obtain a reasonable thermodynamic description of a thermal vacancy in the metastable disordered body centered cubic (bcc_A2) phase, which had been consistently ignored in previous thermodynamic assessments. The present approach was first applied to obtain the thermodynamic descriptions for pure metastable bcc Ni and Zn, and then in the binary Ni-Zn system. The thermodynamic descriptions for both the metastable disordered bcc_A2 phase and the stable ordered bcc_B2 β phases in the Ni-Zn binary system were updated based on the corresponding experimental equilibria. With these updated thermodynamic descriptions, several drawbacks, including the multiple solutions for thermal vacancy concentrations and the artificial phase boundaries in previous assessments, can be avoided. Moreover, the calculated phase boundaries and invariant reactions related to the β phase agree well with the experimental data.

## 1. Introduction

As the simplest but most important type of structural defect, thermal vacancy affects different properties of materials such as diffusivity [1], thermal conductivity [2], and heat capacity [3]. These effects are especially significant in alloys with a body cubic centered (bcc_A2) structure, and the effects are enhanced by increased temperature. Although this defect has typically been ignored in thermodynamic descriptions, it is important to consider the contribution of thermal vacancy in the Gibbs energy of the bcc_A2 phase. In recent studies, positive values like +30*T* [4], +(ln10)*RT* [5,6,7], or +0.2*RT* [8], have been assigned as the Gibbs energy for thermal vacancy in the bcc_A2 phase. Previously, we [9] proposed an effective strategy to determine reasonable thermodynamic descriptions for pure bcc_A2 metals by considering the contribution of thermal vacancy, and successfully applied this approach to bcc W. However, this approach [9] requires further experimental data, such as the concentration of thermal vacancy and heat capacity, especially at temperatures close to melting. Therefore, this strategy is valid only for the stable bcc_A2 phase since it is difficult to experimentally obtain Gibbs values for the metastable bcc_A2 phase of pure metals and alloys. First-principles calculations may be applied to compute thermochemical data based on crystal structural information, but the accuracy of thermodynamic properties calculated close to melting temperatures remains unsatisfactory [10]. Thus, a different approach is needed to effectively estimate Gibbs energy values for the metastable bcc_A2 phase with thermal vacancy.

To describe the Gibbs energy for pure metastable bcc_A2 metals with thermal vacancy, one end-member that contains only thermal vacancies *Va* (*G_Va_*), as well as the interaction parameters between thermal vacancy and pure element *k* (*L_k,Va_*) in metastable bcc_A2 phase, and the end-member for pure element *k*, must be reasonably described. In previous thermodynamic assessments, *G_Va_* was simply set to be 0, resulting in numerical problems during subsequent thermodynamic calculations. For example, using the previous assessment for the Ni-Zn binary system by Xiong et al. [11], the Gibbs energy of the thermal vacancy (*G_Va_*) in the metastable bcc_A2 phase was set to 0, and the interaction parameters *L_Ni,Va_* and *L_Zn,Va_* were, respectively, set to +22,965.2 (in J mole-atom^−1^) and +98,373-0.911*T* (in J mole-atom^−1^, and *T* in Kelvin) to fit the experimental phase boundaries related to the β phase. The β phase is of the CsCl type with a bcc_B2 structure, which is the ordered phase transformed from the metastable bcc_A2 phase. The bcc_A2 phase is commonly associated with disordered solutions. Figure 1 shows the calculated temperature-dependent thermal vacancy concentrations in metastable bcc_A2 Ni and Zn according to the thermodynamic descriptions by Xiong et al. [11]. As shown in Figure 1, there are three analytical solutions for the thermal vacancy concentrations (i.e. ^1^*y_va_*, ^2^*y_va_*, and ^3^*y_va_*) in bcc_A2 Ni and Zn below the corresponding critical temperature points (4620 K for Ni and 1030 K for Zn), where the second derivations of the Gibbs energy for bcc_A2 Ni and Zn both reach 0. The multiple thermal vacancy concentrations at a single temperature result in the non-uniqueness of the molar Gibbs energy, which could cause an unreasonable phase diagram. Moreover, the thermal vacancy concentration of ^3^*y_va_*=1 could cause a pole in the Gibbs energy, which would dramatically stabilize the bcc_A2 phase, resulting in incorrect phase equilibria. In Figure 2, the blue dashed lines indicate the artificial phase boundaries caused by the solution of the unit’s thermal vacancy concentration. Moreover, the Gibbs energy of the metastable disordered bcc_A2 phase may also influence the Gibbs energy of the stable ordered phase. For instance, the ordered β phase in the Ni-Zn system can be described by the order-disorder thermodynamic model [12,13], in which the Gibbs energy for the disordered metastable bcc_A2 phase contributes to the Gibbs energy of the stable ordered β phase. Thus, a more accurate thermodynamic description of the metastable bcc_A2 phase with a thermal vacancy is necessary. 

Consequently, the purpose of this study was (*i*) to develop an effective approach to estimate the Gibbs energy for the metastable bcc_A2 phase with a thermal vacancy, (*ii*) to obtain the thermodynamic descriptions for metastable bcc_A2 Ni and Zn by applying this approach, and (*iii*) to update the thermodynamic descriptions of disordered metastable bcc_A2 and ordered stable β phases in an Ni-Zn binary system by considering the presently obtained thermodynamic descriptions for bcc_A2 Ni and bcc_A2 Zn together with the related phase equilibria. 

## 2. Gibbs Energy for the Metastable bcc_A2 Phase with a Thermal Vacancy

### 2.1. Metastable bcc_A2 Element

Considering a pure element *k*, three phases are assumed to exist over a wide temperature range: the stable reference state, the liquid phase, and the metastable bcc_A2 phase. For instance, the Gibbs energy for the stable reference state and the liquid phase are known from the Scientific Group Thermodata Europe (SGTE) compilation by Dinsdale [14]. To analyze the metastable bcc_A2 pure metals with thermal vacancy, the model *(k*,Va) is employed; the molar Gibbs energy of the bcc_A2 phase ($G_{m}^{bcc\_A2}$) can be expressed as [8,9,15]: (1)$$G_{m}^{bcc\_A2}=G_{k}^{pure-bcc}+\frac{y_{va}}{y_{k}}G_{va}+RT(\ln y_{k}+\frac{y_{va}}{y_{k}}\ln y_{va})+y_{va}L$$
Here, $G_{k}^{pure-bcc}$ is the molar Gibbs energy of pure bcc *k* without a thermal vacancy, and $G_{va}$ is the vacancy energy. The site fractions of *k* and *Va* are $y_{k}$ and $y_{va}$. *L* is the interaction parameter between *k* and *Va*. For the equilibrium state of bcc *k*, one can obtain
(2)$$\frac{\partial G_{m}^{bcc\_A2}}{\partial y_{k}}=-\frac{1}{y_{k}{}^{2}}G_{va}-\frac{1}{y_{k}{}^{2}}RT\ln(1-y_{k})-L=0$$
by applying the constraint $y_{k}+y_{va}=1$. By substituting Equation (2) into Equation (1), the following molar Gibbs energy at the equilibrium state can then be obtained: (3)$$G_{m}^{bcc\_A2}=G_{k}^{pure-bcc}-\frac{{\left(1-y_{k}\right)}^{2}}{y_{k}{}^{2}}G_{va}-\frac{{\left(1-y_{k}\right)}^{2}}{y_{k}{}^{2}}RT\ln(1-y_{k})+RT\ln y_{k}$$

Assuming that $G_{k}^{ref}$ and $G_{k}^{liq}$ are the molar Gibbs energy for pure *k* in the reference state and liquid state, respectively, the lattice stability between bcc_A2 and the reference state of *k* ($\Delta G_{k}^{bcc-ref}=G_{k}^{bcc}-G_{k}^{ref}$) can be derived as
(4)$$\Delta G_{k}^{bcc-ref}=G_{k}^{pure-bcc}-\frac{{\left(1-y_{k}\right)}^{2}}{{\left(y_{k}\right)}^{2}}G_{va}-\frac{{\left(1-y_{k}\right)}^{2}}{{\left(y_{k}\right)}^{2}}RT\ln(1-y_{k})+RT\ln y_{k}-G_{k}^{ref}$$
An equation analogous to Equation (4) can be derived to describe the lattice stability between bcc_A2 and liquid *k* ($\Delta G_{k}^{bcc-liq}=G_{k}^{bcc\_A2}-G_{k}^{liq}$).

To ensure the stabilization of the stable phases (i.e., the reference and liquid states), the lattice stabilities $\Delta G_{k}^{bcc-ref}$ should always be positive below the melting point, and the value of $\Delta G_{k}^{bcc-liq}$ should be positive above the melting point. By solving Equation (4) in the limit case (i.e., when $\Delta G_{k}^{bcc-ref}$ and $\Delta G_{k}^{bcc-liq}$ approach 0), the critical site fraction of *k* ($y_{k}^{crit}$) at each temperature can be obtained, but its analytic solution cannot be unambiguously given. Then, with the obtained critical site fraction $y_{k}^{crit}$, one can deduce the value of the critical interaction parameter $L^{crit}$ at each temperature by applying Equation (2),
(5)$$L_{}^{crit}=-\frac{1}{{\left(y_{k}^{crit}\right)}^{2}}G_{va}-\frac{1}{{\left(y_{k}^{crit}\right)}^{2}}RT\ln(1-y_{k}^{crit})$$

It should be noted that the $L^{crit}$ obtained by using Equation (5) represents a series of points over the entire temperature range. As the interaction parameter in the framework of the CALPHAD (CALculation of PHAse Diagrams) approach is temperature-dependent, one of the simplest ways to obtain the expression of the critical interaction parameter is to employ a related function to fit the points of $L^{crit}$. Additionally, the evaluated critical interaction parameter in Equation (5) is the extreme value, which ensures the stabilization of the stable phases at each temperature. In general, the real interaction parameter should be slightly larger than the obtained critical values.

### 2.2. Analysis of bcc_A2 Alloys

A metastable bcc_A2 alloy including *n* elements with a thermal vacancy is described by the model (*i*, *j*, *k*, …, *n*, *Va*). Next, the molar Gibbs energy of this bcc_A2 alloy is given as
(6)$$\begin{array}{l} G_{m}^{bcc\_A2}=\frac{1}{\sum_{i=1}^{n}y_{i}}(\sum_{i=1}^{n}y_{i}G_{i}^{pure-bcc}+y_{va}G_{va})+\frac{RT}{\sum_{i=1}^{n}y_{i}}(\sum_{i=1}^{n}y_{i}\ln y_{i}+y_{va}\ln y_{va})\\ \;+\frac{1}{\sum_{i=1}^{n}y_{i}}(\sum_{i}y_{i}y_{va}L_{i,va}+\sum_{i}\sum_{j>i}y_{i}y_{j}y_{va}L_{i,j,va}+...)\\ \;+\frac{1}{\sum_{i=1}^{n}y_{i}}(\sum_{i}\sum_{j>i}y_{i}y_{j}L_{i,j}+\sum_{i}\sum_{j>i}\sum_{k>j}y_{i}y_{j}L_{i,j,k}+...) \end{array}$$
where $y_{i}$, $y_{j}$ and $y_{va}$ are the site fractions of element *i*, *j* and a thermal vacancy. $L_{i,va}$ and $L_{i,j,va}$ are the interaction parameters between an element and a thermal vacancy, while $L_{i,j}$ and $L_{i,j,k}$ are the binary and ternary interaction parameters between elements. By applying the relation between the molar fraction and site fraction ($x_{k}=y_{k}/\sum_{i=1}^{n}y_{i}$), Equation (6) can be rewritten as
(7)$$\begin{array}{l} G_{m}^{bcc\_A2}=\sum_{i}x_{i}G_{i}^{pure-bcc}+\frac{y_{va}}{1-y_{va}}G_{va}\\ \;+RT[\sum_{i}x_{i}\ln x_{i}+\ln(1-y_{va})+\frac{y_{va}}{1-y_{va}}\ln y_{va}]\\ \;+\frac{1}{\sum_{i}y_{i}}(\sum_{i}x_{i}y_{va}L_{i,va}+\sum_{i}\sum_{j>i}x_{i}x_{j}y_{va}(1-y_{va})L_{i,j,va}+...)\\ \;+((1-y_{va})\sum_{i}\sum_{j>i}x_{i}x_{j}L_{i,j}+{\left(1-y_{va}\right)}^{2}\sum_{i}\sum_{j>i}\sum_{k>j}y_{i}y_{j}y_{k}L_{i,j,k}+...) \end{array}$$
where $x_{i}$ is the molar fraction of the element *i*. 

As shown in Equation (7), if the concentration of a thermal vacancy approaches 0, the molar Gibbs energy expression will become the normal substitution model.

## 3. Application in Binary Ni-Zn System

### 3.1. Gibbs Energy for Metastable bcc_A2 Ni and Zn

The above method was next applied to metastable bcc_A2 Ni and Zn to estimate the critical interaction parameters. The values for $G_{k}^{ref}$, $G_{k}^{liq}$, and $G_{k}^{pure-bcc}$ in Equation (4) are directly taken from the SGTE database [14] for simplification. The value of $G_{va}$ is fixed as 0.2*RT*, a value slightly larger than the critical value, which ensures a unique equilibrium state as proposed by Frank [8]. A similar treatment was adopted in our previous work to describe a stable bcc W [9]. The points in Figure 3a,b are calculated critical values of the interaction parameter by using Equation (4) in both solid and liquid phase regions of metastable bcc_A2 Ni and Zn at each temperature. These calculated values are critical values to ensure the phase stability of pure Ni and Zn in both liquid and solid phase regions. As shown in Figure 3a,b, the obtained critical values increase with an increase in temperature and show clear linear behavior in both solid and liquid phase regions. We then used the linear temperature equations to fit these points. The fitting interaction parameter expressions for Zn and Ni are presented in Figure 3a,b. The solid lines in Figure 3a,b are calculated by using the fitting interaction parameter expressions in Figure 3a,b; these expressions fit the obtained points. Figure 3c,d show the critical coefficient *L*/*RT* in both the solid and liquid phase regions of pure Ni and Zn. The values of *L*/*RT* increased with the increase of the temperature in the solid phase region and decreased in the liquid phase region. At the melting point, the coefficient reached the maximum value. 

As shown in Figure 3a,b, two separate equations were first used to fit the values of the critical interaction parameters in the solid and liquid phase regions. However, such a piecewise interaction parameter is not common in the CALPHAD approach. Instead, a single equation for the interaction parameter is generally used in the solid to liquid temperature range. Therefore, we simply extend the fitting equation in the liquid phase region to the entire temperature range, which also ensures phase stability in the solid temperature range. As mentioned in Section 2.1, to ensure phase stability, the interaction parameter used in the thermodynamic description should be slightly larger than that of the critical value and then an additional energy +1000 J/mol should be added to the fitted interaction parameters. The interaction parameters between the thermal vacancy and Ni or Zn in bcc_A2 phase are listed in Table 1. Figure 4a shows the calculated molar Gibbs energy of the bcc_A2 phase with a thermal vacancy in pure Ni and Zn compared to the values for the liquid and reference states (i.e. fcc for Ni and hcp for Zn). The Gibbs energy values of the bcc_A2 phase are between the values of the liquid and reference phases for both Ni and Zn over the whole temperature range. Figure 4b presents the molar Gibbs energy values of the bcc_A2 and liquid phases in pure Ni and Zn relative to the reference states and shows that the bcc_A2 phase for both Ni and Zn is metastable over the whole temperature range, which indicates that the calculated Gibbs energy for the bcc_A2 phase can ensure phase stability. For comparison, the relative molar Gibbs energy of the bcc_A2 phase without a thermal vacancy is indicated (black line). The Gibbs energy values of the bcc_A2 phase with a thermal vacancy (red line) approach the values without a thermal vacancy at low temperatures, which may indicate a small influence of thermal vacancies and/or low thermal vacancy concentrations at low temperatures. 

### 3.2. Updated Thermodynamic Descriptions for β Ni-Zn Phase

The Ni-Zn binary system contains six phases: liquid, (Ni), NiZn, γ, β, and NiZn_8_. Several thermodynamic descriptions are available for the Ni-Zn system [11,16,17,18,19,20,21]. However, the stable ordered intermetallic phase β was not described properly using the commonly accepted order-disorder transition model [13,14] in the earlier thermodynamic descriptions [16,17,18,19,20,21]. Xiong et al. [11] re-assessed the Ni-Zn binary system by using a two-sublattice order-disorder transition model to describe the β phase. In that assessment, the Gibbs energy of the vacancy end-member (*G_Va_*) in the bcc_A2 phase was set to 0. This zero vacancy Gibbs energy allowed the stabilization of the metastable bcc_A2 phase, as discussed in Section 1. In this section, the thermodynamic descriptions for the β phase are updated by considering the proposed critical interaction parameters $L_{Ni,Va}^{crit}$ and $L_{Zn,Va}^{crit}$ in the metastable bcc_A2 phase. The assessed results from Xiong et al. [11] reasonably agree with most of the existing experimental data, so the thermodynamic parameters for other phases are directly adopted from that work. 

The metastable disordered bcc_A2 is described with (Ni, Zn, Va)_1_ in an Ni-Zn system, and the stable ordered β phase is molded as (Ni, Zn, Va)_0.5_(Ni, Zn, Va)_0.5_. The Gibbs energy of the ordered β phase ($G_{m}^{\beta }$) can be described by using the order/disorder transitions model [12,13]:(8)$$G_{m}^{\beta }=G_{}^{bcc\_A2}(x_{i})+\Delta G_{}^{order}(y_{i}^{\left(s\right)}),$$
in which $G_{}^{bcc\_A2}$ denotes the Gibbs energy of the disordered bcc_A2 phase, which can be expressed according to Equation (7). $\Delta G_{}^{order}$ is the ordering contribution, including the energy difference of the ordered structure with different site fractions of the ordered or disordered configurations. $y_{i}^{\left(s\right)}$ represents the site fraction of element *i* on the *s*-th sublattice. Detailed descriptions of order/disorder transitions are available in Refs. [12,13]. 

Optimization was performed using the PARROT module in the Thermo-Calc software [22]. Due to the crystallographic equivalence of the two sublattices of the B2 structure, the relations $G_{Ni:Zn}^{\beta }=G_{Zn:Ni}^{\beta }$, $L_{Ni,Va:Ni}^{\beta }=L_{Ni:Ni,Va:}^{\beta }$, $L_{Ni,Va:Zn}^{\beta }=L_{Zn:Ni,Va}^{\beta }$, and $L_{Zn,Va:Ni}^{\beta }=L_{Ni:Zn,Va}^{\beta }$ (*G* is the end-member energy and *L* is the interaction parameter in β phase) are used in the present assessment. Table 1 lists the parameters obtained for the metastable bcc_A2 and stable β phases. 

Figure 5 compares the Gibbs energies of the bcc_A2 phase with and without consideration of thermal vacancies in Ni-Zn alloys at 850 and 1050 °C. The molar Gibbs energy of the bcc_A2 phase with a thermal vacancy shows obvious deviations from the energy without a thermal vacancy. The deviations at 1050 °C are larger than those at 850 °C, indicating increased deviations of Gibbs energy with increasing temperature. This indicates that the effects of thermal vacancies on Gibbs energy are augmented at high temperatures. Figure 6 shows the calculated Ni-Zn binary phase diagram together with the experimental data [23,24,25,26,27,28,29,30] over the entire composition range (a) and in the vicinity of the β phase (b). As can be seen in Figure 6a, the calculated phase equilibria agree very well with all the experimental data available in the literature [23,24,25,26,27,28,29,30]. Moreover, the presently calculated phase boundaries of the β phase in Figure 6b show slightly better agreement with the experimental data than those calculated by Xiong et al. [11] (in dashed lines). Further, the artificial phase boundaries of the metastable bcc_A2 phase have been removed by using the newly obtained thermodynamic parameters. Detailed comparison of the invariant reactions related to the β phase between the present calculations and data in the literature [25,26,28] are listed in Table 2. As shown in Table 2, the presently calculated invariant reaction temperatures and compositions agree well with the experimental data [25,26,28]. The agreement in the Ni-Zn binary system further validates the use of the newly obtained parameters to describe the metastable bcc_A2 phase.

## 4. Conclusion

An effective method to obtain the Gibbs energy of a metastable bcc_A2 phase with thermal vacancy was proposed in the present work, to ensure the phase stability of the Gibbs energy over the entire composition and temperature range.The present approach was applied to obtain the Gibbs energy of metastable bcc_A2 Ni and Zn. The thermodynamic descriptions for the metastable bcc_A2 and the stable ordered β phases in Ni-Zn binary systems were then updated based on the corresponding experimental data. Unreasonable thermal vacancy concentrations and artificial phase boundaries were avoided. Moreover, the calculated phase boundaries in the Ni-Zn system agreed well with most of the experimental data. 

## Figures and Tables

**Figure 1 materials-12-00292-f001:**
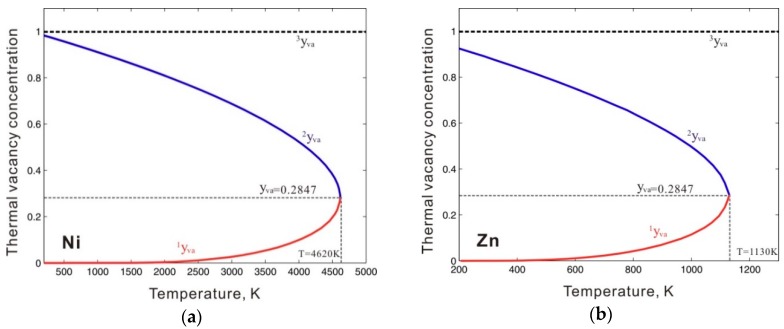
Thermal vacancy concentrations of (**a**) bcc Ni and (**b**) bcc Zn evaluated according to the thermodynamic descriptions by Xiong et al. [11]. ^1^*y_va_*, ^2^*y_va_*, and ^3^*y_va_* are the three analytical solutions for the thermal vacancy concentrations in bcc_A2 Ni or Zn.

**Figure 2 materials-12-00292-f002:**
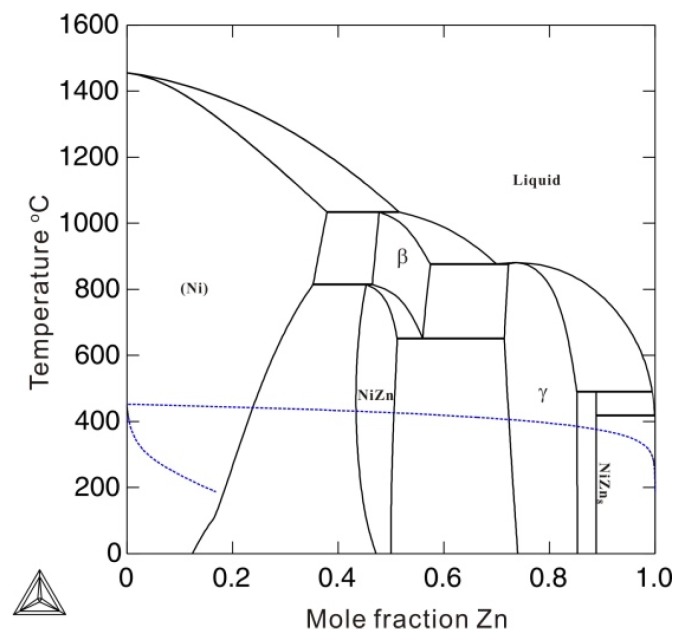
Calculated phase diagram of a binary Ni-Zn system according to the thermodynamic description by Xiong et al. [11]. The blue dashed lines are the artificial phase boundaries of the bcc_A2 phase.

**Figure 3 materials-12-00292-f003:**
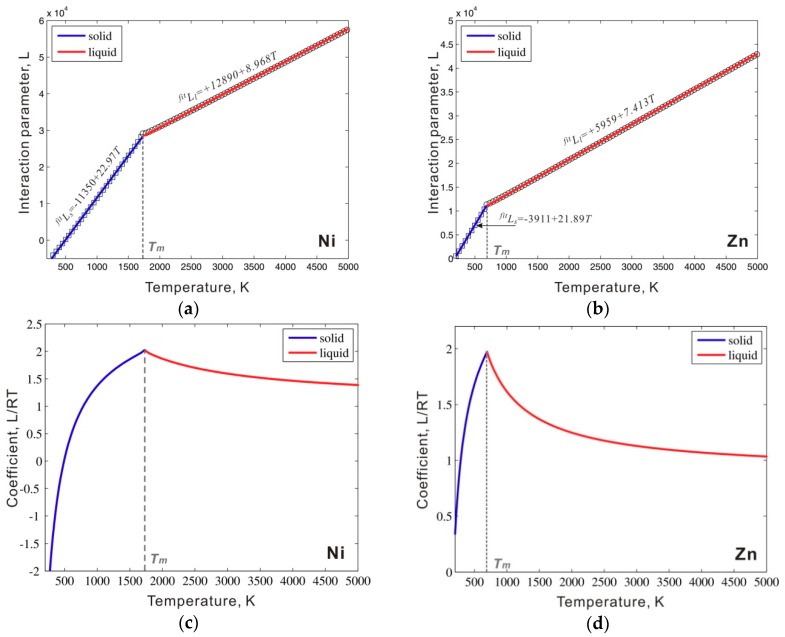
Critical interaction parameters ($L^{crit}$) in the metastable bcc_A2 phase between a thermal vacancy along with the fitted interaction parameters (solid line) for (**a**) Ni and (**b**) Zn; the calculated coefficient *L*/*RT* in (**c**) Ni and (**d**) Zn.

**Figure 4 materials-12-00292-f004:**
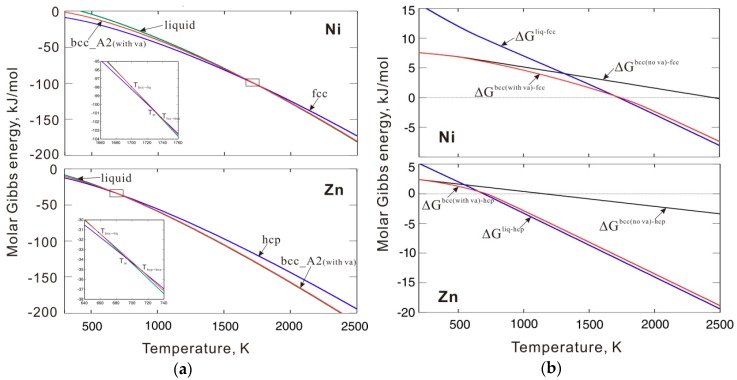
(**a**) Molar Gibbs energy of bcc_A2 with a thermal vacancy along with those of liquid and reference phases for pure Ni and Zn; (**b**) Molar Gibbs energy of bcc_A2 with and without a thermal vacancy and liquid phases for pure Ni and Zn related to the reference states. Here, ΔG denotes differences of Gibbs energy between two phases (for instance $\Delta G^{liq-fcc}=G^{liquid}-G^{fcc}$ ). The reference states are fcc Ni and hcp Zn, respectively.

**Figure 5 materials-12-00292-f005:**
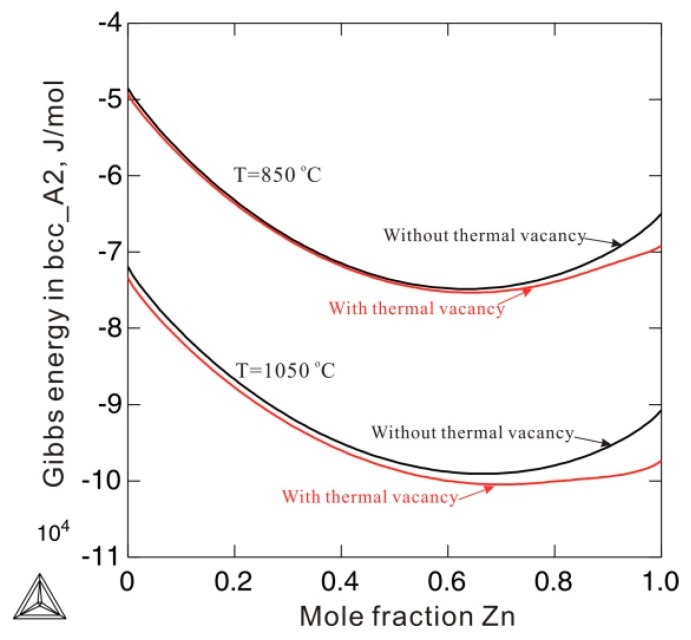
Comparison of molar Gibbs energy of the bcc_A2 phase with (red solid lines) and without a thermal vacancy (black dash lines) in an Ni-Zn binary system at 850 °C and 1050 °C.

**Figure 6 materials-12-00292-f006:**
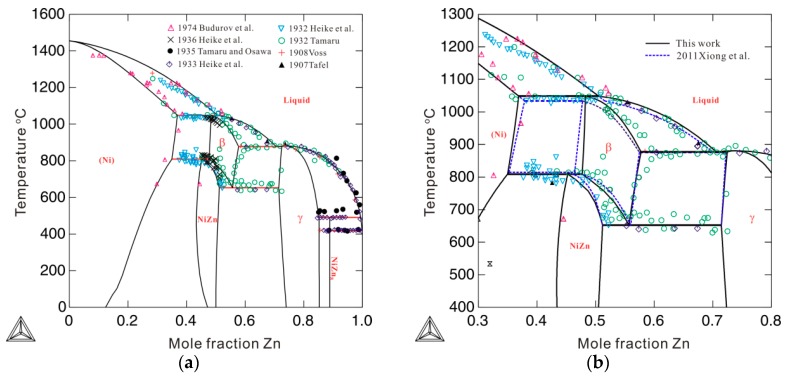
Calculated phase diagram of a binary Ni-Zn system according to the presently obtained thermodynamic descriptions, compared with the experimental data from Refs. [23,24,25,26,27,28,29,30]: (**a**) over the entire composition range and (**b**) from 30 to 80 at.% Zn. The blue dashed lines are the phase boundaries related to the β phase calculated by Xiong et al. [11].

**Table 1 materials-12-00292-t001:** Summary of the optimized parameters for bcc_A2 and β phase in Ni-Zn binary system.

Phases	Parameters (J/mol-atom)
bcc_A2 (metastable) Model: (Ni, Zn,Va)_1_	$\begin{array}{l} G_{Ni}^{bcc\_A2}=+8715.1-3.56T+{}^{o}G_{Ni}^{fcc}\\ G_{Zn}^{bcc\_A2}=+2886.9-2.51T+{}^{o}G_{Zn}^{hcp}\\ G_{Va}^{bcc\_A2}=+0.2RT\\ L_{Ni,Va}^{bcc\_A2}=+13890.0+8.97T\\ L_{Zn,Va}^{bcc\_A2}=+6959.0+7.41T\\ L_{Ni,Zn}^{bcc\_A2}=-80420.8+34.12T \end{array}$
bcc_B2 (β, stable) Model: (Ni, Zn,Va)_0.5_ (Ni, Zn,Va)_0.5_	$\begin{array}{l} G_{Ni:Ni}^{\beta }=G_{Zn:Zn}^{\beta }=+0.0\\ G_{Ni:Va}^{\beta }=G_{Va:Ni}^{\beta }=+0.0\\ G_{Zn:Va}^{\beta }=G_{Va:Zn}^{\beta }=+0.0\\ G_{Ni:Zn}^{\beta }=G_{Zn:Ni}^{\beta }=-5439.5-10.71T\\ L_{Ni,Va:Ni}^{\beta }=L_{Ni:Ni,Va}^{\beta }=+7178.6\\ L_{Ni,Va:Zn}^{\beta }=L_{Zn:Ni,Va}^{\beta }=-23441.0\\ L_{Zn,Va:Zn}^{\beta }=L_{Zn:Zn,Va}^{\beta }=+0.0\\ L_{Zn,Va:Ni}^{\beta }=L_{Ni:Zn,Va}^{\beta }=-20477.8+6.65T \end{array}$

**Table 2 materials-12-00292-t002:** List of the invariant reactions related to the β phase, calculated according to the presently obtained thermodynamic parameters, compared to the literature data [11,25,26,28].

Reactions	Type	Compositions (at.% Zn)			Temperature (°C)	Ref.
Liquid + (Ni) = β	Peritectic	50.5	36.9	48.3	1049	This work (Cal.)
		51.6	37.9	47.8	1034	[11] (Cal.)
		53.3	35.9	49.3	1043	[26] (Exp.)
		52.8	40.4	50.0	1040	[25] (Exp.)
Liquid = β + γ	Eutectic	70.5	57.8	72.5	877	This work (Cal.)
		70.0	57.5	72.3	876	[11] (Cal.)
		69.8	59.9	-	872	[26] (Exp.)
		68.8	58.4	69.8	875	[28] (Exp.)
(Ni) + β = NiZn	Peritectoid	47.7	35.0	45.2	808	This work (Cal.)
		46.5	35.3	45.4	815	[11] (Cal.)
		51.6	35.87	50.0	804	[26] (Exp.)
		47.3	33.8	-	810	[28] (Exp.)
β = NiZn + γ	Eutectoid	55.7	51.2	71.5	652.5	This work (Cal.)
		56.0	51.2	71.5	652	[11] (Cal.)
		54.3	52.2	73.6	675	[26] (Exp.)
		53.3	51.8	74.3	650	[25] (Exp.)

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
