# Peer review of "On Gibbs Energy for the Metastable bcc_A2 Phase with a Thermal Vacancy in Metals and Alloys"

_materials, 2019, doi:10.3390/ma12020292_

Round 1
Reviewer 1 Report
In the manuscript, Ni-Zn binary system has been assessed using CALPHAD approach with implementation of vacancies energy. As I can see authors did significant progress in understanding practically important phase diagram which potentially has high relevance for the community. I only suggest to give in Fig 1 not only modelled diagram but also actual experimental diagram for comparison.
Reviewer 2 Report
Present manuscript describes theoretical investigation of thermodynamics characteristic for metastable bcc Ni and Zn phases and Ni-Zn phase diagrams.
This is a paper of moderate interest and low scientific novelty and may be recommended for a publication only after a major revision as noted below.
The article is long, poorly written and difficult for perceiving, but understandable.
However, there are some serious concerns that need to be addressed:
1) According to the Figure 6 and Table 2 – the calculations in ref. 2 and in the present work are both in good agreement with the existing experimental data. What novelty of your research from previous research work (e.g. from the data in ref 11)?
2) Please rewrite the article title – it should be clear and informative. The same applies to Abstract: sentences should be clear and informative, unnecessary words (e.g. “In this paper, an effective…”, “The present approach”) should be omitted. Please avoid using abbreviations like bcc_A2 in the Abstract.
3) A thorough revision of language and style, preferably by a native speaker, is required. I have noted many deficiencies in language and grammar, for example:
p. 2 line 61 – “What’s more…”
p.6 lines 171-172 – “Here, ΔG detonates Gibbs difference between two phases…”
p. 7 lines 210-211 – “It shows that the values of molar Gibbs energy with and without thermal vacancy show obvious deviations.”
p. 7 lines 214-216 –“The phase boundaries of the ordered bcc_B2 show a better agreement with the experimental data according to the present assessment, compared with the previous assessment”
p. 7 line 219 – “…the presently calculated results agree very well with the experimental information”
The manuscript should be shortened and clarified. All redundant phrases such as “an effective approach”; “…are chosen as the target in the present paper. The major aims are…”; “In this section, the molar Gibbs energy of bcc alloys with thermal vacancy in bcc phase will be extended.”; “In the present work” (many times).
4) I suggest all technical details about program or methods used (e.g. p. 7 lines 198-201 and etc) move into special section (Experimental part or Experimental details).
5) All abbreviations used in the manuscript should be explained. E.g. what does the bcc_A2 and bcc_B2 means? Is it denotes to metastable and stable β bcc phases?
6) The Table 2 should be clarified and reorganized. There are a lot of sub- and super-scripted abbreviations, it is better to simplify and remove them.
7) Figure 6 should be redesigned and improved – please add different colors on the various experimental rows. Some experimental data are missing. For example phase diagrams from ref. 11 (Figure 2) – are much better organized.
Reviewer 3 Report
The manuscript presents a theoretical study aiming at improving the thermal vacancy description within the CALPHAD method. The proposed improvement was applied to Ni-Zn binary system.
The present reviewer, however, fails to see the claimed improvement. The applied method and theory are both classic and readily accessible. Also clearly shown in Figure 6(b), there is only a slight difference between the present work and the work by Xiong et al (2011).
I would highly recommend the authors revise the manuscript that the claimed improvement are more clearly stated and clarified.
Round 2
Reviewer 2 Report
The authors made a great work and significantly improved the readability of manuscript text in the revised version. The colourful versions of the Figures provide a good representation of experimental data. I am satisfied with answers on my questions.
However, some typos due to AutoCorrect are sill present, e.g. p.6 lines 171-172 "ΔG detonates the difference of Gibbs energies between two phases" probably it should be "ΔG denotes the difference of Gibbs energies between two phases".
Please, double-check the manuscript text in order to eliminate such typos.
Finally I recommend the revised version of the manuscript for publication after a minor revision.
